# LongMagpie: A Self-synthesis Method for Generating Large-scale Long-context Instructions

**Chaochen Gao**[1,2], **Xing Wu**[1,2,*] **Zijia Lin**[4], **Debing Zhang**[3], **Songlin Hu**[1,2,*]
[1]Institute of Information Engineering, Chinese Academy of Sciences
[2]School of Cyber Security, University of Chinese Academy of Sciences
[3]Xiaohongshu Inc, [4]Tsinghua University
{gaochaochen,wuxing,husonglin}@iie.ac.cn
dengyang@xiaohongshu.com, linzijia@tsinghua.edu.cn

## Abstract

High-quality long-context instruction data is essential for aligning long-context large language models (LLMs). Despite the public release of models like Qwen and Llama, their long-context instruction data remains proprietary. Human annotation is costly and challenging, while template-based synthesis methods limit scale, diversity, and quality. We introduce LongMagpie, a self-synthesis framework that automatically generates large-scale long-context instruction data. Our key insight is that aligned long-context LLMs, when presented with a document followed by special tokens preceding a user turn, auto-regressively generate contextually relevant queries. By harvesting these document-query pairs and the model's responses, LongMagpie produces high-quality instructions without human effort. Experiments on HELMET, RULER, and Longbench v2 demonstrate that LongMagpie achieves leading performance on long-context tasks while maintaining competitive performance on short-context tasks, establishing it as a simple and effective approach for open, diverse, and scalable long-context instruction data synthesis.

## 1 Introduction

Large Language Models (LLMs) have demonstrated impressive capabilities across a wide range of tasks, with recent advancements significantly extending their context lengths [29, 1, 18]. The ability to process long documents is essential for complex applications such as Longbook QA [7], document summarization [49], and code planning [5]. However, fine-tuning LLMs to leverage long contexts requires access to high-quality long-context instruction data [8, 2]. While the model weights of several open-source LLMs, such as Qwen [54] and Llama [19], have been made publicly available, the corresponding instruction datasets for long-context training remain proprietary. This closed-data paradigm poses a substantial barrier to the advancement of open-source long-context models.

Existing methods for creating open-source instruction data face substantial limitations when extended to long contexts. (1) Human labor costs are prohibitively high for creating diverse, high-quality long-context instruction data. The annotation difficulty is substantially greater than for short-context data, requiring individuals to read documents spanning thousands of tokens before formulating instructions—a demonstrably challenging task. (2) Existing synthetic approaches, often relying on predefined templates [39] or seed questions [47], do not guarantee the diversity needed for effective long-context instruction. While existing projects [26, 52, 2] attempt to broaden seed data diversity, creating large-scale long-context instructions with high quality and diversity remains an expensive and time-consuming process.

---

[*]Corresponding author.

39th Conference on Neural Information Processing Systems (NeurIPS 2025).

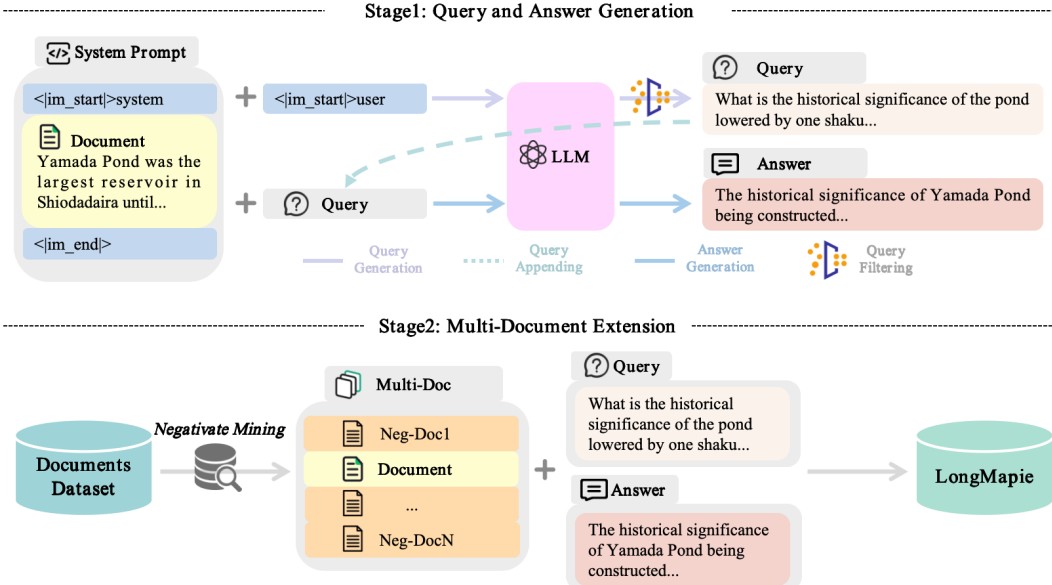

Figure 1: LongMagpie pipeline overview. Stage one: a document serves as a system prompt, a special user token triggers query generation, followed by the model response. Stage two: combines the query-response pair with the source document and sampled documents from the corpus to create challenging multi-document long-instruction data.

A recently proposed self-synthesis method, Magpie [53], has gained widespread attention for eliminating the need for seed instructions and prompt engineering required by previous approaches [47, 26, 52, 2]. It creates alignment data by prompting aligned LLMs with only special tokens preceding a user turn, leveraging their auto-regressive nature. Inspired by Magpie, we introduce **LongMagpie**, a self-synthesis method for generating large-scale long-context instruction data without human annotation or complex prompting. A key observation is that long-context understanding often involves document-based question answering, such as RAG or long document QA. Thus instruction-tuned LLMs such as Qwen [54] and Llama [19] internalize patterns of document-query relationships during their long-context instruction training. Thus, when aligned models are presented with only a document, followed by the special tokens that typically precede a user query, they auto-regressively generate contextually relevant queries about that document. By leveraging this behavior, we can automatically create high-quality instruction-response datasets for long-context training without explicit prompting or manual intervention.

This approach offers advantages: it scales efficiently to generate diverse, high-quality long-context instructions without labor costs or complex prompt engineering; produces naturally varied queries that probe different aspects of documents; and eliminates complex pipeline components required by previous methods. Furthermore, we extend LongMagpie beyond single documents to multi-document contexts, creating more challenging scenarios that require distinguishing relevant information such as RAG [17]. This multi-document extension enhances the model's ability to handle complex real-world applications that frequently involve reasoning across multiple information sources while providing a natural way to increase context length and task difficulty without additional computational overhead.

To further balance the long-context and short-context capabilities, we introduce the $p$-Mix strategy, which addresses the performance degradation on short-context tasks when models are predominantly trained on long-context instructions. This strategy employs a probabilistic mixing approach that begins by prepending a short-context instruction to each training sequence, followed by a dynamic sequence constructed through probabilistic sampling. Specifically, with probability $P_L$, a long-context instruction (generated by LongMagpie) is appended; otherwise, with probability $1 - P_L$, another short-context instruction is selected. This process continues iteratively until approaching the maximum sequence length $L_{max}$. $p$-Mix effectively prevents the model from overfitting to long-context patterns while maintaining strong performance across diverse task scenarios.

Through extensive evaluation on HELMET [55], RULER [22], and Longbench v2 [4] benchmarks, we demonstrate that models trained on LongMagpie-generated data achieve leading performance. When incorporated with $p$-Mix, our approach maintains competitive performance on short-context tasks. We conduct detailed analytical experiments on the LongMagpie method to explain its effectiveness. The positive experimental results demonstrate that LongMagpie represents a meaningful step toward democratizing long-context capabilities for LLMs, making high-quality long-context instruction accessible to the broader research community.

Our main contributions are:

- We introduce a novel self-synthesis approach for generating high-quality long-context instruction data that leverages the auto-regressive nature of aligned LLMs, eliminating the need for human annotation or predefined examples.
- We propose the $p$-Mix technique, a probabilistic mixing strategy that effectively balances the model's performance on both long-context and short-context tasks.
- We conduct extensive evaluations demonstrating that models trained on LongMagpie-generated data achieve leading results on long-context benchmarks compared to existing methods.
- We provide in-depth analyses revealing the key factors contributing to LongMagpie's effectiveness, including query diversity and quality.

## 2 Method

This section introduces LongMagpie, our method for synthesizing long-context instruction data. We first describe the key insight of our approach, followed by the detailed pipeline of LongMagpie and $p$-Mix strategy for balancing long-context and short-context capabilities.

### 2.1 Key Insight: Auto-Regressive Document-Query Generation

The foundation of LongMagpie is a key observation about aligned long-context LLMs: when provided with a document followed by tokens that typically precede a user query (without the query itself), these models generate contextually relevant queries about that document. This behavior stems from the fact that long-context understanding often involves document-based question answering tasks such as RAG and long document QA. During instruction tuning, models like Qwen and Llama internalize document-query relationship patterns, enabling them to auto-regressively predict meaningful questions when presented with document-only contexts. This capability allows us to synthesize diverse, high-quality instruction data without human annotation, predefined templates, or seed questions.

Formally, for an aligned LLM $\mathcal{M}$ with vocabulary $\mathcal{V}$, we define the document-query generation process as follows: given a document $D = \{d_1, d_2, ..., d_n\} \in \mathcal{V}^n$ and pre-query template $T_{pre} = \{t_1, t_2, ..., t_m\} \in \mathcal{V}^m$ (containing tokens indicating a user or query role, e.g., `<|im_start|>user`), we provide input $X = D \oplus T_{pre}$, where $\oplus$ denotes sequence concatenation. The model then generates a sequence $Q = \{q_1, q_2, ..., q_k\} \in \mathcal{V}^k$ representing a query related to document $D$. This process can be described as:

$$p_{\mathcal{M}}(Q \mid D, T_{\text{pre}}) = \prod_{i=1}^{k} p_{\mathcal{M}}(q_i \mid D, T_{\text{pre}}, q_{<i}),$$
(1)

This approach differs fundamentally from traditional prompt engineering or instruction-following, as we are not explicitly instructing the model to generate a query about the document. Instead, we leverage the model's learned patterns of document-query relationships that emerge from its instruction training.

### 2.2 LongMagpie Pipeline

The LongMagpie pipeline consists of two main steps: (1) query and answer generation, and (2) extension to a multi-document setting.

### 2.2.1 Query and Answer Generation

**Document Preparation** We collect diverse documents from various domains and lengths, primarily using curated resources like Fineweb. These documents span domains including science, history, literature, and technical topics, with an average length of approximately 1.6k tokens in our primary dataset. This provides a range of context lengths while focusing on truly long-context scenarios.

**Query Generation** For each document $D$, we construct an input sequence $X = D \oplus T_{pre}$, where $T_{pre}$ contains tokens preceding a user query in the model's instruction template. For example, the tokens for Llama-3-Instruct model are `<|start_header_id|>user` and for Qwen-2.5-Instruct are `<|im_start|>user`. We pass $X$ to the aligned LLM and sample a completion $Q$ until an end-of-template token is generated or a maximum length is reached. This completion represents a contextually relevant query. By generating multiple queries per document with different sampling parameters, we create diverse document-query pairs that naturally vary in complexity.

**Response Generation** For each document-query pair $(D, Q)$, we construct a standard instruction prompt by combining the document, query, and tokens that precede an assistant response (e.g., `<|eot_id|><|start_header_id|>assistant<|end_header_id|>` for Llama-3-Instruct). We then generate a response $R$, forming a complete instruction triplet $(D, Q, R)$ for long-context training. If the same model is used for both query and response generation, these steps can be consolidated without manual intervention.

**Query Filtering** In query generation, we observed that LLMs occasionally continue the input document rather than generate queries, particularly when the model size is small. To ensure the quality of the generated queries, we applied two filtering strategies: (1) **Rule-based filtering**: we retain queries that end with a question mark as a simple heuristic to identify interrogative sentences; (2) **Length-based filtering**: we discard generated texts longer than 1.5k characters, as they are typically descriptive passages rather than valid queries.

### 2.2.2 Multi-Document Extension

To enhance task diversity and real-world applicability, we extend LongMagpie to multi-document settings. Many tasks require reasoning over several related documents rather than a single one. Our approach involves:

- Obtaining $x$ documents $\{D_1, \ldots, D_x\}$ as negative documents via random sampling, where $x$ is drawn uniformly from 0 to $n$ (with $n = 0$ reducing to the standard single-document QA setting). .
- Concatenating documents using a special separator token (e.g., `<|doc_sep|>`) to form $D_{\text{multi}} = D_1 \oplus$ `<|doc_sep|>` $\oplus \cdots \oplus D_x$.
- Generating queries and responses as in the single-document pipeline, producing triples $(D_{\text{multi}}, Q, R)$ requiring cross-document reasoning.

### 2.3 *p*-Mix: Balancing Long-Context and Short-Context Capabilities

Fine-tuning predominantly on long-context data degrades performance on short-instruction tasks [2, 52]. To balance these capabilities, we introduce *p*-Mix, a novel instruction data hybridization strategy. The core idea is twofold. First, to emulate the typical non-contextual start of general tasks, we sample a short-context instruction at the beginning of each training sequence. Second, we append subsequent data segments probabilistically to construct a mixed-context sequence up to length $L_{max}$. With probability $P_L$, a long-context instruction (generated by LongMagpie) is chosen; otherwise, with probability $1 - P_L$, another short-context sample is chosen. This process repeats until approaching the target sequence length, ensuring each instance starts with a short, context-free instruction followed by a dynamically mixed sequence of long and short segments. This prepares the model for diverse real-world scenarios. The procedure is formalized in Algorithm 1, and we conduct an ablation study of the parameters related to *p*-Mix in Appendix A.9.

# 3  Experiments

In this section, we describe our experimental setup, present our main results, and analyze the factors that contribute to LongMagpie's performance.

## 3.1  Experimental Setup

**Dataset Generation**  Using the LongMagpie pipeline described in Section 1, we generate a long-context instruction dataset using Qwen2.5-70B-Instruct, with documents sampled from FineWeb-Edu [34]. FineWeb-Edu is a subset of the FineWeb dataset, comprising 1.3 trillion tokens extracted from educational web content.

**Compared Datasets**  We compare LongMagpie-generated data against several widely used instruction datasets. These include datasets specifically designed for long contexts and standard short-context datasets adapted for long-context fine-tuning based on ProLong [16].

- **Long Instruction Datasets** We compare with two long-context datasets: **ChatQA** [52] combines multiple data sources, including LongAlpaca12k [8] and GPT-4 samples from Open Orca [28], containing 1.5 million synthetic instructions. In this work, we refer to ChatQA2 as ChatQA by default; **LongAlign** [2] generates questions and answers for long documents by prompting LLMs.
- **Short Instruction Datasets** Following findings that concatenated short instructions benefit long-context capabilities [16], we include: **Tulu** [24], an open-source collection based on Llama 3.1; **Magpie** [53], a self-synthesis method using template prefixes; and **UltraChat** [11], comprising 1.5 million multi-turn dialogues. We concatenate samples from these datasets to reach the target context length during fine-tuning.

### 3.1.1  Model Training

We select `Llama-3-8B-NExtLong-512K-Base` [15] as our base model, which has undergone extensive long-context continued pre-training. The batch size is 4M tokens for 250 steps, a total of 1B tokens for baseline datasets and LongMagpie. The same training configuration is applied across all datasets to ensure a fair comparison. Further details are provided in Appendix A.1.

### 3.1.2  Evaluation Benchmarks

**Long-context Evaluation**  We evaluate our models on three comprehensive long-context benchmarks. These benchmarks provide a holistic assessment of models' abilities to utilize long contexts effectively across different tasks and complexity levels.

- **HELMET** [55] evaluates long-context models across diverse application-centered tasks with context lengths up to 128k tokens, using model-based evaluation that prioritizes complex tasks for better real-world performance prediction.
- **RULER** [22] provides fine-grained evaluation of long-context reasoning with synthetic tasks that offer flexible control over sequence length and complexity to identify performance bottlenecks beyond simple retrieval.
- **LongBench-v2** [4], an upgrade to LongBench [3], assesses extremely long-context understanding (8k to 2M words) through 503 expert-validated questions across six categories, revealing a need for improved ultra-long reasoning capabilities.

**Short-context Evaluation**  To further evaluate the model's ability to follow short instructions, we select 7 widely-used short-context datasets: HellaSwag (Hel.) [57], Lambada_OpenAI (Lam.) [35], ARC-Challenge (AR-C.) [9], ARC-Easy (AR-E.), PIQA [6], WinoGrande (Win.) [38], and Logiqa (Log.) [30].

## 3.2  Main Results

As shown in Table 1, models trained solely on **LongMagpie** data already set a leading performance on long-context evaluation, topping HELMET (62.10), RULER (91.17), LongBench-v2 (34.4) and the LongAVG score (62.56) within the *Long Instruction Data* group. The performance gains are

Table 1: Main experimental results comparing LongMagpie with other methods on long-context and short-context benchmarks. Best scores in each column are bolded. LongAVG is the average of HELMET, RULER, and Longbench v2, ShortAVG is the average of different short-context tasks.

| Dataset | Long Evaluation | | | | Short Evaluation |
|---|---|---|---|---|---|
| | HELMET | RULER | Longbench v2 | LongAVG | ShortAVG |
| **Short Instruction Data** | | | | | |
| Tulu | 61.93 | 87.92 | 28.4 | 59.42 | 63.90 |
| Magpie | 60.18 | 87.06 | 31.4 | 59.55 | 63.32 |
| UltraChat | 60.55 | 83.85 | 30.4 | 58.27 | **64.43** |
| **Long Instruction Data** | | | | | |
| ChatQA | 60.23 | 89.82 | 30.8 | 60.28 | 63.58 |
| LongAlign | 57.79 | 86.08 | 24.5 | 56.12 | 60.97 |
| LongMagpie | **62.10** | **91.17** | **34.4** | **62.56** | 62.37 |
| ***p*-Mix: Long + Short Instruction Data** | | | | | |
| ChatQA + UltraChat | 60.80 | 87.42 | 31.4 | 59.87 | 64.38 |
| LongAlign + UltraChat | 60.98 | 89.49 | 30.6 | 60.36 | 64.17 |
| LongMagpie + UltraChat | **62.11** | **89.70** | **33** | **61.60** | 64.10 |

substantial compared to existing long-context instruction datasets: LongMagpie outperforms ChatQA by +1.87 on HELMET, +1.35 on RULER, and +3.6 on LongBench-v2, yielding a +2.28 improvement on LongAVG. The gap is even more pronounced when compared with LongAlign, where LongMagpie delivers gains of +4.31 on HELMET, +5.09 on RULER, and +9.9 on LongBench-v2, resulting in a remarkable +6.44 improvement on LongAVG. The strong performance of LongMagpie on long-context tasks demonstrates the effectiveness of our self-synthesis approach for generating high-quality long-context instruction data without human annotation or seed examples.

Among the models trained with *p*-Mix strategy, which mixes LongMagpie with other short-instruction datasets, **LongMagpie + UltraChat** achieves the best or tied-best scores on HELMET (62.11), RULER (89.70) and LongAVG (61.60) among *all* mixed datasets. It also retains a competitive Short-AVG accuracy (64.10), only 0.33 below the overall best, confirming that 1) The long-context signals produced by our self-synthesis method are highly complementary to existing short-instruction data, and 2) The probabilistic mixing schedule effectively balances these two instruction regimes, yielding models that are robust across both ultra-long reasoning and everyday short-instruction scenarios. These results highlight the practical value of *p*-Mix: it preserves the strength of LongMagpie on long-context tasks while simultaneously mitigating the typical performance drop on short-context benchmarks. We provide further analysis to demonstrate the advantages of *p*-Mix compared to alternative mixing strategies in Section 4.2.

## 4 Ablation Studies

This section first analyzes the key configurations that influence LongMagpie's performance, then evaluates the quality and diversity of its generated queries, and finally assesses the its resource efficiency.

### 4.1 Impact of Different Multi-Document Settings

To increase instruction difficulty and further enhance the model's ability to capture long-range dependencies, we introduce a multi-document setting. With a certain probability, the document associated with a generated query-answer pair is mixed with $x$ randomly sampled documents from the corpus, where $x$ is drawn uniformly from 0 to $n$ (with $n = 0$ reducing to the standard single-document QA setting). Table 2 provides the detailed performance scores for different values of $n$ in the multi-document setting, corresponding to the trends shown in Appendix A.8. We observe that the multi-document strategy significantly improves performance on long-context tasks (from 60.19 to 62.56). As the value of $n$ increases, the performance on long-context tasks improves and degrades, with the best performance observed when $n = 10$. We hypothesize that this trend is due to

Table 2: Detailed results for the impact of the maximum number of documents ($n$) in a user prompt.

| $n$ | HELMET | RULER | Longbench v2 | LongAVG | ShortAVG |
|---|---|---|---|---|---|
| 0 | 60.13 | 89.04 | 31.4 | 60.19 | **63.20** |
| 5 | 61.42 | 89.91 | 31.4 | 60.91 | 61.98 |
| 10 | **62.10** | **91.17** | **34.4** | **62.56** | 62.37 |
| 20 | 61.75 | 91.08 | 32.8 | 61.88 | 62.04 |
| 40 | 62.08 | 90.77 | 31.0 | 61.28 | 62.37 |
| 80 | 61.15 | 90.65 | 31.0 | 60.93 | 62.13 |

Table 3: $p$-Mix better balances the performance of long-context and short-context than other mixing strategies.

| Strategy | HELMET | RULER | Longbench v2 | LongAVG | ShortAVG |
|---|---|---|---|---|---|
| No Mix | 62.10 | **91.17** | **34.4** | **62.56** | 62.37 |
| Sequential Mix | 61.60 | 88.85 | 31.8 | 60.75 | 61.89 |
| Simple Mix | 61.84 | 89.65 | 31.2 | 60.90 | 64.04 |
| $p$-Mix (Ours) | **62.11** | 89.70 | 33.0 | 61.60 | **64.10** |

an excessive number of documents increasing the task difficulty beyond the model's learning capacity, thereby leading to a drop in performance.

## 4.2 Impact of Different Mixing Strategy

To investigate the effectiveness of the $p$-Mix Strategy, we compare $p$-Mix with three alternative mixing approaches: (1) No Mix: training solely on LongMagpie data without short-context SFT datasets; (2) Sequential Mix: first training on short-context data (UltraChat) then fine-tuning on long-context data (LongMagpie), similar to [11]; (3) Simple Mix: directly combining and shuffling long and short data in a single training stage, similar to approaches used with LongAlign [2]; and (4) $p$-Mix (Ours): our proposed strategy from Algorithm 1 that pre-pends short instructions and probabilistically mixes segments. As Table 3 demonstrates, alternative strategies struggle to balance long-context and short-context performance compared to our $p$-Mix approach. In contrast, our $p$-Mix strategy demonstrates a superior balance: it achieves a competitive LongAVG of 61.60 (notably better than sequential and simple mixing, and only a slight trade-off compared to no mixing) while attaining the best ShortAVG score of 64.10. This highlights the efficacy of the $p$-Mix approach in maintaining strong long-context reasoning abilities while significantly bolstering performance on short, non-contextual tasks. More details can be found in Appendix A.9.

## 4.3 Impact of Different Data Size

To investigate the impact of data volume on model performance, we train our models using two different sizes of LongMagpie-generated data: 190k and 450k samples. As shown in Table 4, scaling up the training data from 190k to 450k samples leads to consistent improvements across all long-context evaluation benchmarks. Specifically, we observe gains of +0.81 on HELMET, +0.52 on RULER, and +1.8 on Longbench v2, resulting in a +1.05 improvement in the overall LongAVG metric. This demonstrates that increasing the volume of high-quality long-context instruction data significantly enhances the model's ability to comprehend and reason over extended contexts.

## 4.4 Impact of Different Source Model Size

To assess the impact of different models on data synthesis, we use LongMagpie to generate two 450k long-context instructions respectively by the Qwen-2.5-7B model and the Qwen-2.5-70B model. As shown in Table 5, using the larger 70B model improves LongAVG performance (59.61 $\rightarrow$ 62.56), and shows similar performance on ShortAVG. This superior performance likely stems from larger models' enhanced ability to model long-context capabilities [50], which translates to better results when applied to the LongMagpie method.

Table 4: Increasing the volume of training data improves performance on long-context benchmarks.

| Source Model | Data Volume | HELMET | RULER | Longbench v2 | LongAVG | ShortAVG |
|---|---|---|---|---|---|---|
| Qwen-2.5-70B | 190k | 61.29 | 90.65 | 32.6 | 61.51 | 62.30 |
| Qwen-2.5-70B | 450k | **62.10** | **91.17** | **34.4** | **62.56** | **62.37** |

Table 5: Using the larger source model improves performance on long-context benchmarks..

| Source Model | Data Volume | HELMET | RULER | Longbench v2 | LongAVG | ShortAVG |
|---|---|---|---|---|---|---|
| Qwen-2.5-7B | 450k | 59.28 | 86.95 | 32.6 | 59.61 | 62.18 |
| Qwen-2.5-70B | 450k | **62.10** | **91.17** | **34.4** | **62.56** | **62.37** |

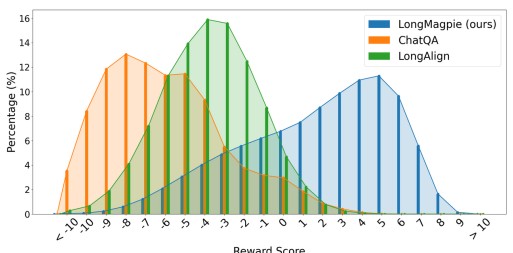 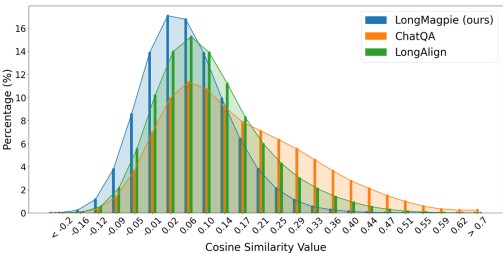

(a) Reward model scores for different datasets.     (b) Query similarities within different datasets.

Figure 2: Analysis of LongMagpie-generated data quality and diversity. (a) higher reward model scores indicates higher quality. (b) lower pairwise query similarity indicates better diversity.

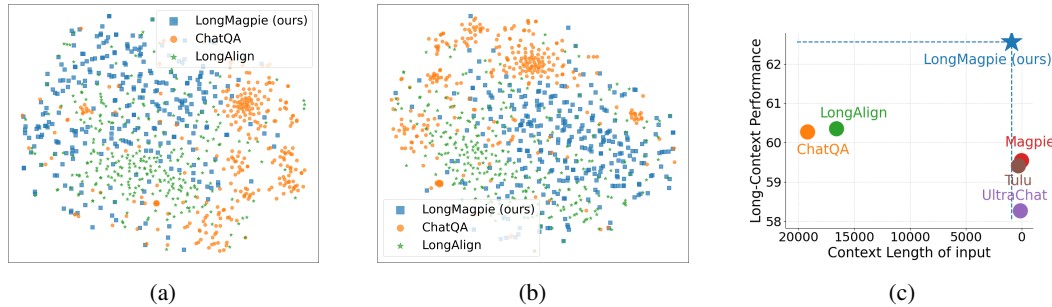

(a)           (b)           (c)

Figure 3: Visualizations of LongMagpie characteristics: (a,b) t-SNE visualizations of query embeddings from different datasets showing LongMagpie's dispersed distribution indicating diversity; (c) Long-context performance vs. token consumption showing LongMagpie's superior performance.

## 4.5  Analysis of of LongMagpie Queries

**Higher Quality of LongMagpie Queries**  We use the Reward Model FsfairX-Llama3-RM-v0.1 [12] to score three long-context fine-tuning datasets. As shown in Figure 2a, the x-axis represents the scores given by the reward model, and the y-axis represents the proportion of data within each dataset corresponding to that score. The overall data quality of LongMagpie is significantly higher than that of ChatQA and LongAlign.

**Better Diversity of LongMagpie Queries**  To investigate the diversity of different datasets, we sampled 300 queries from each dataset, inferred their embeddings using the jina-embeddings-v3 [42] model, and visualized their distribution using t-SNE [45], as shown in Figure 3. It can be observed that LongMagpie's distribution is more dispersed, reflecting its better diversity.

Furthermore, we repeated the following experiment 30 times: sampled queries from each dataset, calculated the pairwise similarity between the sampled queries within each dataset, and aggregated the distributions of all similarities, as shown in Figure 2b. It can be seen that LongMagpie queries generally exhibit lower similarity among themselves, which also reflects their good diversity.

### 4.6 Sample Efficiency of LongMagpie

We analyze the sample efficiency of various long-context instruction synthesis methods by quantifying the average token processing requirements during instruction synthesis. As illustrated in Figure 3c, LongMagpie exhibits exceptional sample efficiency, achieving superior long-context performance while processing substantially fewer tokens per instruction (averaging 1.6K tokens)[2]. This efficiency stands in stark contrast to methods like ChatQA and LongAlign, which consume 10-13× more tokens per instruction during synthesis yet produce inferior performance outcomes. LongMagpie's remarkable sample efficiency facilitates greater scalability and diversity.

### 4.7 Sample-Count-Controlled Comparison

To ensure a sample-count-controlled comparison, we train on 190k samples across different methods. For ChatQA, we use its original 190k dataset; for LongAlign, we follow its original construction strategy to generate a 190k version. Results are shown in Table 6.

Table 6: Comparison under equal data size (190k samples).

| Method | Data Size | HELMET | RULER | LongBenchV2 |
|--------|-----------|--------|-------|-------------|
| ChatQA | 190k | 60.23 | *89.82* | 30.8 |
| LongAlign | 190k | *60.63* | 87.36 | **33.0** |
| LongMagpie | 190k | **61.29** | **90.65** | *32.6* |

We further scale up the LongAlign dataset to 450k samples to compare its scalability. Results are shown in Table 7.

Table 7: Scalability comparison with increased data size (450k samples).

| Method | Data Size | HELMET | RULER | LongBenchV2 |
|--------|-----------|--------|-------|-------------|
| LongAlign | 450k | 60.62 | 88.77 | 33.2 |
| LongMagpie | 450k | **62.10** | **91.17** | **34.4** |

As shown in Table 6 and Table 7, LongMagpie consistently outperforms LongAlign on average, especially as the data scale increases. We attribute this to its ability to generate more diverse and higher-quality questions (as illustrated in Figure 2) through adaptive query generation, rather than relying on fixed prompt templates or seed questions.

Moreover, prior methods often depend on domain-specific long-context data or long-context-capable LLMs, which hinders their scalability. For example, ChatQA synthesizes data using NarrativeQA and needs to be combined with LongAlpaca12k and OpenOrca to reach 190k samples. LongAlign requires long documents and long-context models for data synthesis, and also needs to be mixed with short-text instruction data. In contrast, LongMagpie uses only general short-document datasets (around 1.6k tokens on average, as shown in Figure 3c) and a simple, scalable method, enabling efficient synthesis at scale without external instruction data.

## 5 Related Work

### 5.1 Long-Context Data Synthesis

Existing approaches to synthesizing long-context data can be divided into two categories.

**Continuation-Oriented Methods** Approaches in this category generate long-context data by concatenating shorter documents. Early methods [37, 8] used random sampling and concatenation, but failed to maintain meaningful long-range dependencies. Later approaches preserved semantic coherence through document clustering [20] or nearest-neighbor retrieval [40]. Quest [14] balances

---

[2]Our multi-document extension approach enables arbitrary context length extension without incurring additional computational overhead.

relevance and diversity using keyword matching. NExtLong [15] decomposes a document into multiple meta-chunks and extends the context by interleaving hard negative distractors retrieved from pretraining corpora. However, these methods focus on pre-training rather than instruction tuning. In contrast, LongMagpie directly generates instruction-following data with the model's auto-regressive capabilities.

**Instruction-Oriented Methods**   There exist many approaches to generate long-context instruction data [59, 46, 23, 43]. Representative works include WildLong [26] uses templates and seed questions, LongAlign [2] employs Self-Instruct with packing strategies but requires curated examples, ChatQA [31] blends QA datasets with conversational QA, ChatQA 2 [52] packs documents into 32-128K token contexts, LOGO [44] adapts self-synthesis for long-context alignment, and GATEAU [41] focuses on valuable instruction selection. These methods obtain high-quality data through complex pipelines. In contrast, LongMagpie eliminates seed questions, and complex pipelines by leveraging aligned LLMs' ability to generate contextually relevant queries when provided only with documents.

### 5.2   Synthesis Methods for Short-Context Instruction Data

Recent studies scale synthesis across various dimensions: Unnatural Instructions [21] yields diverse instructions through paraphrasing; WizardLM [51] uses evolutionary strategies for challenging variants; GLAN [25] eliminates templates by generating tasks from taxonomies; BARE [58] improves factual correctness; and Humpback [27] performs instruction back-translation. Domain-specific approaches like MetaMath [56] generate specialized content. Magpie [53] demonstrates aligned LLMs can autoregressively generate diverse instructions without human annotation or seed examples. Motivated by Magpie, LongMagpie extends this paradigm to long-context settings by leveraging document-query relationship patterns from instruction tuning, enabling diverse long-context instruction data without specialized prompting.

## 6   Conclusion

This paper introduces LongMagpie, a self-synthesis method that automatically generates large-scale long-context instruction data without human annotation or seed examples. Extensive experiments on HELMET, RULER, and Longbench v2 demonstrate that models trained on LongMagpie data achieve leading performance on long-context tasks while maintaining competitive short-context capabilities when combined with our proposed $p$-Mix strategy. This work establishes LongMagpie as an effective approach for democratizing long-context capabilities.

## 7   Limitations

First, LongMagpie unavoidably inherits biases from the source instruction-tuned LLMs, which future work should detect and mitigate. Second, the current implementation of LongMagpie inadequately covers tasks requiring long-form outputs, as it primarily focuses on document-query relationships rather than extended reasoning or generation. Future research should expand support for diverse output formats and complex analytical tasks.

## Acknowledgement

This work is supported by the National Natural Science Foundation of China (No. U24A20335).

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

# A  Detailed Experimental Results

## A.1  Training Config

We employ the AdamW [33] optimizer with parameters $\beta_1 = 0.9$ and $\beta_2 = 0.95$. Following ProLong [16], we concatenate samples up to 64K sequence length and apply the document masking technique to prevent interactions between independent sequences. Additionally, we utilize FlashAttention-2 [10] and ZeRO [36] to optimize memory usage and accelerate training. The detailed training config is shown in Table 8.

Table 8: Model Training Configuration.

|  | training setting |
| --- | --- |
| Initial Model | Llama-3-8B-NExtLong-512K-Base |
| rotary-emb-base | 128,000,000 |
| $\beta_1$ | 0.9 |
| $\beta_2$ | 0.95 |
| lr | $2e^{-5}$ |
| precision | bfloat16 |
| gradient-clipping | 1.0 |
| weight-decay | 0.1 |
| lr-decay-style | cosine |
| train-iters | 250 |
| seq-length | 65536 |
| GPU-type | H100 |
| GPU-numbers | 8 |
| training-time | 10h |

## A.2  Detailed Results of HELMET

We present results across a comprehensive suite of HELMET tasks, including Recall, RAG, ICL, Re-rank, LongQA, Cite, Summ, and RULER. The complete evaluation results are shown in Table 9. In Section 3, we report the average performance excluding the Cite and Summ tasks, as these two are newly included and evaluated in the latest version of our experiments.

Table 9: Evaluation results across HELMET tasks.

| Method | Recall | RAG | ICL | Re-rank | LongQA | Cite | Summ. | RULER |
| --- | --- | --- | --- | --- | --- | --- | --- | --- |
| ChatQA | 93.34 | **66.47** | 80.36 | 23.74 | **37.25** | 15.18 | 20.61 | 89.82 |
| LongAlign | 92.43 | 59.05 | 81.20 | 27.12 | 29.14 | 17.93 | 24.32 | 86.08 |
| LongMagpie | **97.53** | 63.37 | **85.84** | **28.60** | 35.16 | **19.99** | **26.36** | **91.17** |

## A.3  Detailed Results of LongBench v2

We further evaluate our approach on the LongBench V2 benchmark, which measures multi-domain long-context understanding across a variety of tasks, including multi-document QA (Multi-Doc QA), long in-context learning (ICL), single-document QA (Single-Doc QA), code repo understanding (Code), long-dialogue history understanding (Long-dial.), and long structured data understanding (Long Stru.). The detailed results are shown in Table 10. Our proposed method (**LongMagpie**) consistently outperforms prior approaches across most categories, showing powerful performance on long dialogue history understanding and multi-document question answering.

Table 10: Evaluation results across LongBench v2 tasks.

| Method | Multi-Doc QA | ICL | Single-Doc QA | Code | Long-dial. | Long Stru. |
| --- | --- | --- | --- | --- | --- | --- |
| ChatQA | 25.6 | 34.57 | 36.0 | 24.0 | 25.64 | 30.30 |
| LongAlign | 20.0 | 27.16 | 28.57 | 24.0 | 17.95 | 21.21 |
| LongMagpie | **28.8** | **35.8** | **37.14** | **28.0** | **46.15** | **33.33** |

## A.4 Distribution of Generated Query Types

We categorize the generated QA pairs into various task types. As shown in Table 11, our framework already generates a substantial number of instances beyond traditional document-query pairs, including tasks related to summarization and complex structured extraction.

Table 11: Distribution of generated QA pair types.

| Category | Count |
|---|---|
| Precise Retrieval | 201,306 |
| Summarization | 91,118 |
| Advice Seeking | 51,609 |
| Planning or Reasoning (Multi-step Analysis) | 38,526 |
| Comparative or Choice-Based Task | 25,342 |
| Math or Data Analysis | 8,679 |
| Complex Structured Extraction | 5,725 |
| Creative Task | 3,339 |
| Coding & Debugging | 2,999 |

In addition, the generated task types in LongMagpie often align closely with document content—e.g., code documents yield code-related queries, and structured texts lead to extraction tasks. We will expand to more diverse domains and formats to broaden task coverage, with concrete examples to be included in the future.

## A.5 Effect of Retrieval-Focused Training Data

Based on our experimental results, we find that retrieval-focused training data do not limit the model's generalization to other long-context skills. On the contrary, the improved retrieval capability facilitates performance across various tasks. Intuitively, effective retrieval is a foundational skill for handling long-context inputs, as models must first identify relevant information before generating accurate responses. The importance of retrieval in long-context models has also been widely recognized in prior work [48, 13].

To investigate this more directly, we conduct additional experiments on the 190k-sample dataset. Specifically, we vary the proportion of Precise Retrieval data while adjusting the other data distributions accordingly. One setting reduces the Precise Retrieval portion from 50% to 30%, and the other increases it to 70%.

Table 12: Performance comparison under different proportions of Precise Retrieval data.

| Precise Retrieval (%) | Recall | RAG | ICL | Re-rank | LongQA | Cite | Summ. | RULER |
|---|---|---|---|---|---|---|---|---|
| 30% | 97.63 | 62.99 | 80.92 | 25.44 | 34.72 | 19.30 | **26.12** | 90.71 |
| 50% | 97.29 | 62.72 | **85.12** | 26.26 | 35.05 | 20.39 | 24.32 | 90.65 |
| 70% | **98.85** | **63.38** | 84.16 | **26.85** | **36.26** | **22.02** | 24.86 | **90.93** |

As shown in Table 12, increasing the proportion of Precise Retrieval data improves the model's Recall performance, which also leads to consistent gains in downstream tasks such as RAG, Re-rank, LongQA, and Cite, confirming that retrieval-centric training benefits general long-context capabilities.

## A.6 Replacing Ultrachat with Magpie

We conduct experiments using the combination of LongMagpie and Magpie. The results were roughly comparable to the LongMagpie + Ultrachat setting. As shown in Table 13, we observe a slight improvement in long-context performance, while the performance on short-context tasks decreased slightly.

## A.7 Safety Analysis

We performed a safety analysis using the Llama-Guard-3-8B [32] model to classify the generated content. As shown in Table 14, the resulting dataset is overwhelmingly safe, with 99.86% of samples

Table 13: Comparison between LongMagpie + Ultrachat and LongMagpie + Magpie.

| Method | HELMET | RUELR | LongBenchV2 | LongAVG | ShortAVG | LongAVG + ShortAVG |
|---|---|---|---|---|---|---|
| LongMagpie + Ultrachat | **62.11** | 89.70 | 33.00 | 61.60 | **64.10** | **62.85** |
| LongMagpie + Magpie | 61.95 | **90.47** | **33.40** | **61.94** | 63.17 | 62.56 |

Table 14: Safety classification results of the LongMagpie dataset.

| Category | Percentage (%) |
|---|---|
| Safe | 99.8603 |
| Specialized Advice | 0.1147 |
| Intellectual Property | 0.0072 |
| Non-Violent Crimes | 0.0063 |
| Hate | 0.0033 |
| Indiscriminate Weapons | 0.0028 |
| Child Sexual Exploitation | 0.0009 |
| Violent Crimes | 0.0009 |
| Defamation | 0.0009 |
| Elections | 0.0002 |
| Sexual Content | 0.0007 |
| Code Interpreter Abuse | 0.0007 |
| Privacy | 0.0005 |
| Sex-Related Crimes | 0.0002 |
| Suicide & Self-Harm | 0.0002 |

categorized as safe. This suggests that our pipeline can produce high-quality instructional data with minimal safety concerns.

## A.8   Impact of Multi-Document Setting

Figure 4 illustrates the performance variation under different multi-document configurations.

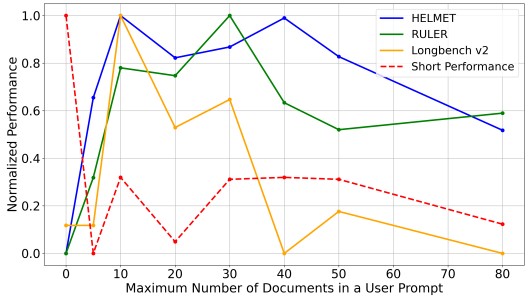

Figure 4: Impact of the multi-document setting on model performance. As the number of documents increases, the performance on long-context tasks improves and then decreases.

## A.9   Ablation Study on *p*-Mix Strategy Parameters

To further understand the behavior of the *p*-Mix strategy, we conducted an ablation study on its key parameters: the number of initial short-context samples pre-pended ($N_S$), and the probability ($P_L$) of selecting a long-context sample during the probabilistic mixing phase (see Algorithm 1). The results, presented in Table 15, showcase how different configurations impact overall performance on both long and short tasks evaluation benchmarks. These experiments were conducted with $n = 10$ for the multi-document context length parameter.

Table 15: Detailed ablation results for different parameter settings of the *p*-Mix strategy. $N_S$ is the number of pre-pended short tasks. $P_L$ is the long-context selection probability.

| $N_S$ | $P_L$ | HELMET | RULER | Longbench | LongAVG | ShortAVG |
|---|---|---|---|---|---|---|
| 0 | 0.2 | 61.38 | 88.52 | 29.60 | 59.83 | 64.17 |
| | 0.4 | 61.84 | 89.65 | 31.20 | 60.90 | 64.04 |
| | 0.6 | 61.64 | 90.51 | 31.00 | 61.05 | 63.92 |
| | 0.8 | 61.48 | 90.54 | 30.40 | 60.81 | 63.41 |
| 1 | 0.2 | 61.62 | 88.05 | 31.60 | 60.42 | **64.39** |
| | 0.4 | 62.11 | 89.70 | 33.00 | **61.60** | **64.10** |
| | 0.6 | 61.74 | 90.58 | 29.80 | 60.71 | 63.71 |
| | 0.8 | 61.45 | 90.66 | 28.80 | 60.30 | 63.33 |
| 5 | 0.2 | 61.41 | 88.12 | 29.80 | 59.78 | 64.16 |
| | 0.4 | 61.70 | 88.67 | 31.20 | 60.52 | 64.13 |
| | 0.6 | 61.90 | 90.07 | 30.00 | 60.66 | 63.97 |
| | 0.8 | 61.34 | 90.53 | 31.00 | 60.96 | 63.68 |
| 30 | 0.2 | 61.17 | 85.67 | 31.80 | 59.55 | **64.41** |
| | 0.4 | 60.77 | 85.30 | 30.00 | 58.69 | 64.25 |
| | 0.6 | 60.67 | 86.09 | 30.80 | 59.19 | **64.39** |
| | 0.8 | 60.60 | 84.42 | 30.00 | 58.34 | 64.21 |

---

**Algorithm 1** Hybrid SFT Data Construction with short-context Pre-pending and Probabilistic Mixing

---

1: **procedure** CONSTRUCTHYBRIDSAMPLE($D_S, D_L, P_L, L_{max}, sep$)
2:  Initialize $S_{concat} \leftarrow$ empty sequence  ▷ $D_S$: set of short-context SFT samples, $D_L$: set of long-context SFT samples  ▷ $P_L$: probability of selecting a long-context sample, $L_{max}$: max sequence length  ▷ $sep$: separator token/sequence between samples
3:  $s_0 \leftarrow$ RandomSample($D_S$)
4:  $S_{concat} \leftarrow$ FormatSample($s_0$)
5:  $current\_length \leftarrow$ Length($S_{concat}$)
6:  **while** $current\_length < L_{max}$ **do**
7:   $rand \leftarrow$ RandomReal($0, 1$)
8:   **if** $rand < P_L$ **then**  ▷ Select long-context sample with probability $P_L$
9:    $l_{next} \leftarrow$ RandomSample($D_L$)
10:    $formatted\_l_{next} \leftarrow$ FormatSample($l_{next}$)
11:    **if** $current\_length +$ Length($sep$) $+$ Length($formatted\_l_{next}$) $\leq L_{max}$ **then**
12:     $S_{concat} \leftarrow S_{concat} \oplus sep \oplus formatted\_l_{next}$
13:     $current\_length \leftarrow$ Length($S_{concat}$)
14:    **else**
15:     **break**  ▷ Next sample exceeds $L_{max}$
16:    **end if**
17:   **else**  ▷ Select short-context sample with probability $1 - P_L$
18:    $s_{next} \leftarrow$ RandomSample($D_S$)
19:    $formatted\_s_{next} \leftarrow$ FormatSample($s_{next}$)
20:    **if** $current\_length +$ Length($sep$) $+$ Length($formatted\_s_{next}$) $\leq L_{max}$ **then**
21:     $S_{concat} \leftarrow S_{concat} \oplus sep \oplus formatted\_s_{next}$
22:     $current\_length \leftarrow$ Length($S_{concat}$)
23:    **else**
24:     **break**  ▷ Next sample exceeds $L_{max}$
25:    **end if**
26:   **end if**
27:  **end while**
28:  **return** $S_{concat}$
29: **end procedure**

---

