# OpenReview forum: "LongMagpie: A Self-synthesis Method for Generating Large-scale Long-context Instructions"
_NeurIPS.cc/2025/Conference — NeurIPS 2025 poster_

### Official Review · Reviewer_KGZd · 2025-06-17

**Clarity:** 3
**Significance:** 3
**Originality:** 3
**Rating:** 4
**Confidence:** 4

**Summary:**

LongMaggie introduces an automated framework for generating large-scale long-context instruction data without human annotation or predefined templates. The core insight is that aligned LLMs (e.g., Qwen, Llama) auto-regressively generate contextually relevant queries when provided with a document followed by special user-role tokens. The pipeline involves: (1) Extracting diverse documents (e.g., from FineWeb-Edu), (2) Generating queries via LLMs, (3) Filtering low-quality queries (rule/length-based), (4) Producing responses to form instruction triplets. The method extends to multi-document settings for complex reasoning and proposes p-Mix—a probabilistic data-mixing strategy—to mitigate short-context performance degradation. Evaluations on HELMET, RULER, and LongBench-v2 show state-of-the-art long-context performance. Models trained solely on LongMaggie data outperform competitors, while *p*-Mix hybrids maintain competitive short-context results.

**Questions:**

1. For query generation failure cases (Sec 2.2.1), how frequently do smaller models output non-query text? Could a classifier improve filtering beyond rule/length heuristics?

2. Beyond QA, can LongMaggie support tasks requiring long-form outputs by modifying the generation protocol?

**Ethical Concerns:**

["NO or VERY MINOR ethics concerns only"]

**Final Justification:**

Thanks for the authors' response. I have read it and decided to maintain my positive score.

**Limitations:**

1. Architectural Bias: Synthesized data inherits limitations of source LLMs (Qwen/Llama), risking error propagation.

2. Narrow Output Scope: Emphasis on document-query pairs overlooks tasks needing extended reasoning (e.g., multi-step analysis).

3. Computational Cost: Heavy reliance on 70B models for high-quality data generation raises accessibility barriers.

4. Ethical Gaps: No discussion of misuse potential (e.g., generating deceptive long-context content) or fairness audits.

**Quality:**

3

**Strengths And Weaknesses:**

Strengths:

1. Novelty: Leverages inherent LLM capabilities for query synthesis, eliminating costly human annotation/seed examples.

2. Effectiveness: Achieves SOTA on major long-context benchmarks (HELMET: 62.10, RULER: 91.17).

3. Scalability: Generates 450K high-quality instructions efficiently (1.6K tokens/sample vs. 10-13× for competitors).

Weaknesses:

1. Bias Inheritance: Acknowledges uncorrected biases from source LLMs.

---

> ### Author Rebuttal · Authors · 2025-07-31
>
> We sincerely appreciate your response! Your detailed and insightful feedback plays a crucial role in improving our article. The following text further clarifies some questions.
>
> ---
>
> **Q1: "For query generation failure cases (Sec 2.2.1), how frequently do smaller models output non-query text? "**
>
>
> **A1:** When using the smaller Qwen2.5-7B-Instruct model to generate data, the success rate is 77.2%. This means that 22.8% of the generated texts are non-query text.
>
> **Q2: "Could a classifier improve filtering beyond rule/length heuristics?"**
>
> **A2:**
> To compare the performance of a classifier and rule/length heuristics, we sample 1,000 outputs and use the Qwen2.5-72B-Instruct model as the classifier for comparison with rule/length heuristics. The prompt provided to Qwen2.5-72B-Instruct is as follows:
> ```
> Classify the text below strictly as either QUERY or NON-QUERY.
> QUERY: Represents a direct question or explicit search intent (e.g., "What is...", "How to...", "Find restaurants near...").
> NON-QUERY: Represents statements, descriptions, continuations, or any non-interrogative content.
> ```
>
> Ultimately, only 48 outputs show inconsistent classifications between the two methods, yielding an agreement rate of **95.2%**. This suggests that the differences between the two approaches have minimal impact.
>
>
>
> **Q3: "Beyond QA, can LongMaggie support tasks requiring long-form outputs by modifying the generation protocol?"**
>
>
> **A3:** The core functionality of LongMagpie lies in its ability to generate diverse, document-adaptive queries. We categorize the generated queries into the following types:
>
> | Category                                | Count   |
> |-----------------------------------------|---------|
> | Precise Retrieval                       | 201,306 |
> | Summarization                           | 91,118  |
> | Advice Seeking                          | 51,609  |
> | Planning or Reasoning (Multi-step Analysis) | 38,526  |
> | Comparative or Choice-Based Task        | 25,342  |
> | Math or Data Analysis                   | 8,679   |
> | Complex Structured Extraction           | 5,725   |
> | Creative Task                           | 3,339   |
> | Coding & Debugging                      | 2,999   |
>
> These queries support a wide range of downstream tasks, particularly those requiring long-form responses. For example, Planning or Reasoning tasks involving multi-step analysis can be addressed by models such as DeepSeek-R1, while Creative Tasks are well-suited for models optimized for long-form content generation.
>
>
> **Q4: "Architectural Bias: Synthesized data inherits limitations of source LLMs (Qwen/Llama), risking error propagation."**
>
> **A4:** We plan to adopt the following strategies to detect and mitigate biases inherited from the source LLMs:
>
> * **Cross-model verification**: Inspired by the multi-agent debate framework proposed by Google [1], we will generate QA pairs for each document using multiple LLMs, then conduct a debate among them. A separate judge model is used to evaluate the debated outputs and select the most appropriate responses. This process helps reduce the likelihood of propagating biases or errors from any single model.
>
> * **Quality filtering**: We employ FsfairX-Llama3-RM-v0.1 to score and filter out low-quality generated data. As shown in Figure 2, this filtering step effectively improves the overall quality of the dataset.
>
> * **Safety filtering**: In addition, we perform safety checks using Llama-Guard-3-8B. Our analysis shows that 99.86% of the data passes the safety criteria. Detailed safety statistics are provided in **A7**.
>
> **Q5: "Narrow Output Scope: Emphasis on document-query pairs overlooks tasks needing extended reasoning (e.g., multi-step analysis)."**
>
> **A5:** We categorize the query tasks in **A3**, and our dataset already includes instances that involve extended reasoning (Planning or Reasoning), such as multi-step analysis.
>
>
> **Q6: "Computational Cost: Heavy reliance on 70B models for high-quality data generation raises accessibility barriers."**
>
> **A6:**  Considering the computational cost, we will release our data in the revised version to make it more accessible to the community. We hope this will further promote the development of long-context models in the field.
>
>
> **Q7: "Ethical Gaps: No discussion of misuse potential (e.g., generating deceptive long-context content) or fairness audits."**
>
> **A7:** We performed a safety analysis using the Llama-Guard-3-8B model to classify the generated content. The results are summarized below:
>
> | Category                    | Percentage (%) |
> |-----------------------------|----------------|
> | Safe                        | 99.8603        |
> | Specialized Advice          | 0.1147         |
> | Intellectual Property       | 0.0072         |
> | Non-Violent Crimes          | 0.0063         |
> | Hate                        | 0.0033         |
> | Indiscriminate Weapons      | 0.0028         |
> | Child Sexual Exploitation   | 0.0009         |
> | Violent Crimes              | 0.0009         |
> | Defamation                  | 0.0009         |
> | Elections                   | 0.0002         |
> | Sexual Content              | 0.0007         |
> | Code Interpreter Abuse      | 0.0007         |
> | Privacy                     | 0.0005         |
> | Sex-Related Crimes          | 0.0002         |
> | Suicide & Self-Harm         | 0.0002         |
>
> Thanks to the safety alignment of the Qwen model used in generation, the resulting dataset is overwhelmingly safe, with **99.86%** of samples categorized as safe. This suggests that our pipeline can produce high-quality instructional data with minimal safety concerns.
>
>
> ---
>
> [1] Du, Yilun, et al. Improving factuality and reasoning in language models through multiagent debate. ICML 2023.

---

> > ### Comment · Reviewer_KGZd · 2025-08-04
> >
> > Thanks for the reply, I have read and decided to keep my positive score.

---

### Official Review · Reviewer_vum6 · 2025-06-26

**Clarity:** 3
**Significance:** 3
**Originality:** 3
**Rating:** 5
**Confidence:** 3

**Summary:**

The paper presents LongMagpie, a straightforward strategy for sampling long-context instructions from an aligned LLM. This approach extends the original Magpie method, where a language model is given a conditional system prompt along with special tokens and tasked with generating both a user query and its corresponding response. LongMagpie follows a similar setup but introduces documents into the prompt, encouraging the model to produce queries that focus on long-document understanding. The results indicate that the generated instructions effectively capture long-context phenomena, achieving state-of-the-art performance on long-context benchmarks. Furthermore, the authors propose a method for mixing long- and short-context instructions to better balance the model’s performance across both settings.

**Questions:**

- In the data mixing experiments, why was the combination of LongMagpie + Magpie not explored? Since both methods are closely related, it would be interesting to see whether combining them yields better or more balanced performance.

**Ethical Concerns:**

["NO or VERY MINOR ethics concerns only"]

**Final Justification:**

During the rebuttal the authors clarified some of the points raised in my review, however, the justification of the p-mix approach is still limited. I decided to keep my score, which was already high.

**Limitations:**

While the authors acknowledge several limitations, one important aspect is not addressed: the proposed method relies on access to a collection of high-quality documents to generate the long-context instructions. This requirement may restrict the applicability of the approach in low-resource languages or specialized domains where such document collections are scarce or difficult to obtain.

**Quality:**

3

**Strengths And Weaknesses:**

### Strengths
- The paper is well written and easy to follow.
- The main contribution is simple yet effective and potentially impactful.
- The authors conduct several ablations and insightful analyses.

### Weaknesses
- The effectiveness of the second contribution, p-Mix, is unclear. If the goal is to achieve a balanced performance across both long- and short-context tasks, one might consider using the harmonic mean of the ShortAVG and LongAVG scores as a more representative metric. Based on this, the improvement of p-Mix (62.82) over No-Mix (62.46) appears marginal, making it difficult to draw strong conclusions about its impact. While p-Mix clearly shifts the trade-off between long and short performance, it is not evident that it offers a meaningful balance overall.
- There are some concerns regarding the evaluation framework of the main contribution:
  - What model was used to generate the instructions for the (short-context) Magpie dataset? If it differs from the model used for LongMagpie, what is the rationale behind that choice? For a fair comparison of the techniques, it would be more robust to keep the model architecture consistent across datasets.
  - Table 4 highlights that the size of the dataset has a measurable effect on performance. However, the paper does not clearly specify how many data instances are present in the other evaluated datasets. How do their sizes compare to LongMagpie? A clearer and more detailed description of the experimental setup would improve the paper’s transparency and reproducibility.

---

> ### Author Rebuttal · Authors · 2025-07-31
>
> We sincerely appreciate your response! Your detailed and insightful feedback plays a crucial role in improving our article. The following text further clarifies some questions.
>
> ---
>
> **Q1: "...While p-Mix clearly shifts the trade-off between long and short performance, it is not evident that it offers a meaningful balance overall."**
>
>
> **A1:** The p-Mix strategy offers a broader option that caters to scenarios where stronger performance on short-text inputs is required. As shown in Table 3, among all mix approaches, p-Mix achieves the best short-text performance while incurring the least degradation in long-context performance. For use cases that prioritize long-context understanding, we recommend directly employing our best-performing model trained solely on LongMagpie data.
>
>
> **Q2: "What model was used to generate the instructions for the (short-context) Magpie dataset?"**
>
> **A2:**  The instructions for the Magpie dataset were also generated using Qwen2.5-72B-Instruct, which is the same model we used in our work.
>
>
> **Q3: "Table 4 highlights that the size of the dataset has a measurable effect on performance. However, the paper does not clearly specify how many data instances are present in the other evaluated datasets. How do their sizes compare to LongMagpie? A clearer and more detailed description of the experimental setup would improve the paper’s transparency and reproducibility."**
>
> **A3:** To ensure a sample-count-controlled comparison, we train on 190k samples across different methods. For ChatQA, we use its original 190k dataset;  For LongAlign, we follow its original construction strategy to generate a 190k version. Results are shown below:
>
> | Method      | Data Size | HELMET | RUELR | LongbenchV2 |
> |:-----------:|:------------:|:------------:|:------------:|:------------:|
> | ChatQA   | 190k      | 60.23  | _89.82_ | 30.8        |
> | LongAlign   | 190k      | _60.63_  | 87.36  | **33**        |
> | Longmagpie  | 190k      | **61.29**   | **90.65** | _32.6_        |
>
> We further scale up the LongAlign dataset to 450k samples to compare its scalability. Results are as follows:
>
> | Method      | Data Size | HELMET | RUELR | LongbenchV2 |
> |:-----------:|:------------:|:------------:|:------------:|:------------:|
> | LongAlign   | 450k      | 60.62   | 88.77 | 33.2        |
> | Longmagpie  | 450k      | **62.1**   | **91.17** | **34.4**        |
>
> As shown, LongMagpie consistently outperforms LongAlign on average, especially as the data scale increases. We attribute this to its ability to generate more diverse and higher-quality questions (as shown in Figure 2) through adaptive query generation, rather than relying on fixed prompt templates or seed questions.
>
> In addition, prior methods often depend on domain-specific long-context data or long-context-capable LLMs, which hinders their scalability. For example, ChatQA synthesizes data using NarrativeQA and needs to be combined with LongAlpaca12k and OpenOrca to reach 190k samples. LongAlign requires long documents and long-context models for data synthesis, and also needs to be mixed with short-text instruction data. In contrast, LongMagpie uses only general short-document datasets (~1.6k tokens average, as shown in Figure 3) and a simple, scalable method, enabling efficient synthesis at scale without external instruction data.
>
>
>
>
>
>
>
> **Q4: "In the data mixing experiments, why was the combination of LongMagpie + Magpie not explored? "**
>
> **A4:** We conduct experiments using the combination of LongMagpie and Magpie. The results were roughly comparable to the LongMagpie + Ultrachat setting. Specifically, we observed a slight improvement in long-context performance, while the performance on short-context tasks decreased slightly.
>
> | Method      | HELMET | RUELR | LongbenchV2 | LongAVG | ShortAVG | LongAVG + ShortAVG |
> |:-----------:|:------------:|:------------:|:------------:|:------------:|:------------:|:------------:|
> | Longmagpie + Ultrachat  | **62.11**  | 89.70 | 33  | 61.60| **64.10** | **62.85** |
> | Longmagpie + Magpie | 61.95    | **90.47**  | **33.4** | **61.94** |  63.17 | 62.56 |
>
>
> **Q5: "...This requirement may restrict the applicability of the approach in low-resource languages or specialized domains where such document collections are scarce or difficult to obtain."**
>
> **A5:** For low-resource languages or specialized domains where document collections are scarce or hard to obtain, one potential solution is first to use large language models (LLMs) to synthesize domain-relevant documents, followed by employing LongMagpie to generate QA datasets. This approach can partially alleviate the issue. We believe that exploring the synthesis of documents remains an exciting and valuable direction for future research.

---

> > ### Comment · Reviewer_vum6 · 2025-08-04
> >
> > Thank you for your response, I think adding some of these points to the experimental setup will make the paper more clear. I will keep my score.

---

### Official Review · Reviewer_iG7h · 2025-06-30

**Clarity:** 3
**Significance:** 2
**Originality:** 2
**Rating:** 4
**Confidence:** 4

**Summary:**

This paper addresses the need for high-quality, large-scale long-context instruction data for LLMs.
Building on the Magpie framework, which generates queries from domain-specific prompts using pretrained LLMs, LongMagpie replaces these prompts with long context documents. This allows aligned LLMs to self-synthesize relevant document-query-response triplets by simply being presented with a documen, an empty user prompt and special tokens indicating start of users' prompt.

The method extends to multi-document contexts for more complex reasoning and introduces the "p-Mix" strategy to blend long and short-context instructions during training, preventing performance degradation on short-context tasks.

Experiments show models trained on LongMagpie data achieve better performance on long-context benchmarks (HELMET, RULER, LongBench-v2) compared with existing short or long-context instruction-tuning dataset, while p-Mix ensures competitive short-context capabilities. The generated data is proven to be of higher diversity with sample efficiency.

**Questions:**

1. While LongMagpie shows leading performance, some gains over strong baselines appear marginal. Also, data scaling yields modest improvements (+1.05 LongAVG from 190k to 450k samples). Could authors elaborate on the practical significance of these gains, perhaps highlighting scenarios or metrics where the advantage is more pronounced? Is it possible to do further scaling to observe more significant gains?
2. Could the author provide a more granular breakdown of performance for benchmarks like RULER and HELMET across different context length ranges and tasks. This would clarify effectiveness at truly extreme versus moderately long contexts, and the effectivenes across tasks?
3. Training uses equal total tokens (1B), but distinct sample counts may vary between datasets. Since SFT is sensitive to unique examples, not just token count, could authors clarify this comparison or discuss if a sample-count-controlled comparison was considered?
4. How do authors plan to detect and mitigate biases inherited from the source LLMs in the generated data? Additionally, can the framework be extended beyond document-query relationships to cover other critical long-context tasks (e.g., summarization, complex structured extraction, multi-turn dialogue)?
5. Any safely analysis of the generated dataset? It seems important to do filtering when synthesizing data using pretrained LMs.

**Ethical Concerns:**

["NO or VERY MINOR ethics concerns only"]

**Final Justification:**

Author addressed most of my concerns during rebuttal, and I appreciate the new results. Therefore, I decided to raise the rating to 4. However, three concerns still remain.

1. The method inherits bias from the underlying pretrained LLMs.
2. Building on the Magpie framework, which generates queries from domain-specific prompts using pretrained LLMs, LongMagpie replaces these prompts with long context documents. The novelty of this method seems somewhat modest to me.
3. Modest improvement is achieved by scaling from 190k data to 450k data as noted in W4.

**Limitations:**

The authors discussed the limitation adequately.

**Quality:**

2

**Strengths And Weaknesses:**

Strengths
1. The paper effectively tackles the challenge of acquiring large-scale, high-quality long-context instruction data, which has been a major bottleneck for training powerful long-context LLMs.
2. LongMagpie's core innovation lies in its "self-synthesis" approach, which leverages the inherent auto-regressive capabilities of aligned LLMs to generate contextually relevant queries from documents. This method eliminates the need for expensive human annotation or complex, pre-defined templates.
3. The paper provides evidence that the synthetically generated queries are of greater diversity, which are crucial factors for effective instruction tuning.

Weaknesses
1. A fundamental limitation is that LongMagpie, by design, inherits biases from the source instruction-tuned LLMs used for data generation. If the base LLM has biases, these will likely be propagated into the synthetic instruction data.
2. The current method primarily focuses on document-query-response triplets, which might not adequately cover tasks requiring more complex or open-ended outputs. The emphasis is heavily on information extraction and question-answering from documents. Analysis regarding the covered tasks is needed. Additionally, authors can provide concrete examples of the synthesized data so that readers can understand it more clearly.
3. While overall performance is leading, looking at Table 1, some of the improvements over strong baselines (LongMagpie + UltraChat vs. LongAlign + UltraChat) on benchmarks like RULER and HELMET can appear somewhat marginal (e.g., gains of 1 points). This raises questions about the practical significance of some of the reported advances.
4. Although the framework is designed to generate large amounts of data, the ablation study on data size shows that the performance improvement (e.g., +1.05 in LongAVG when scaling from 190k to 450k samples) can be rather modest. This raises a concern about diminishing returns when further scaling up the dataset, suggesting that the framework's scalability in terms of performance gain per additional data unit might be limited beyond a certain point.
5. The paper presents aggregated scores for benchmarks like RULER and HELMET. A more detailed breakdown of performance across different context length ranges and tasks within these benchmarks would be highly beneficial. This would provide a clearer understanding of the method's effectiveness at truly extreme long contexts versus moderately long ones, and its effectiveness across different task types.
6. The authors state that training involves a total of 1B tokens for both baseline datasets and LongMagpie, ensuring comparable computational cost. However, the number of distinct training instances (samples) for these 1B tokens can vary significantly between datasets. Since SFT performance can be heavily influenced by the diversity and quantity of unique examples, not just total token count, this discrepancy could lead to an unfair comparison, potentially disadvantaging baselines with fewer but longer instances. This aspect could be better clarified or controlled for in the experimental design. For example, LongAlign only contains around 10k instances, which is significanly less than those used in LongMagpie.

---

> ### Author Rebuttal · Authors · 2025-07-31
>
> We sincerely appreciate your response! Your detailed and insightful feedback plays a crucial role in improving our article.
>
> ---
>
> **Q1: "some of the improvements over strong baselines (LongMagpie + UltraChat vs. LongAlign + UltraChat) on benchmarks like RULER and HELMET can appear somewhat marginal (e.g., gains of 1 points)..."**
>
> **A1:** We would like to clarify that the best long-context performance is achieved using only LongMagpie data. The p-Mix strategy offers a broader option that caters to scenarios where stronger performance on short-text inputs is required. This design choice, to some extent, comes at the cost of slightly reduced performance on long-context benchmarks. However, for long-context tasks specifically, using only the LongMagpie-generated data yields the best results. Compared to other methods, we obtain the highest scores on HELMET (62.10), RULER (91.17), and LongBench V2 (34.4).
>
>
> **Q2: "...Could authors elaborate on the practical significance of these gains, perhaps highlighting scenarios or metrics where the advantage is more pronounced?"**
>
>
> **A2:** Following the partitioning method in [1], we provide metrics for different sub-tasks.
>
> HELMET and RULER Results:
>
> | Method      | Recall  | RAG     | ICL     | Re-rank | LongQA  | RULER|
> |:-----------:|:------------:|:------------:|:------------:|:------------:|:------------:|:------------:|
> | ChatQA      | 93.34 | **66.47** | 80.36 | 23.74   | **37.25** | _89.82_ |
> | LongAlign   | 92.43   | 59.05   | 81.20   | 27.12 | 29.14   | 86.08 |
> | LongAlign + UltraChat   | _94.81_ | _63.81_   | _84.44_ | **30.48** | 31.36   | 89.49 |
> | LongMagpie  | **97.53** | 63.37 | **85.84** | _28.60_ | _35.16_ | **91.17** |
>
> LongBench V2 Results:
>
> | Method      | Multi-Doc QA | ICL     | Single-Doc QA | Code    | Long-dialogue History Understanding | Long Structured Data Understanding |
> |:-----------:|:------------:|:------------:|:------------:|:------------:|:------------:|:------------:|
> | ChatQA      | _25.6_        | _34.57_ | _36_           | 24      | 25.64                             | 30.3                             |
> | LongAlign   | 20            | 27.16   | 28.57          | _24_      | 17.95                                | 21.21                              |
> | LongAlign + Ultrachat    | 23.2          | 23.46   | 34.29          | 18     | _30.77_                                | **45.45**                          |
> | LongMagpie  | **28.8**      | **35.8**| **37.14**      | **28**  | **46.15**                            | _33.33_                          |
>
>
> These results highlight that LongMagpie consistently achieves either the best or second-best performance across the majority of subtasks, leading to its overall superior performance. In particular, it demonstrates significant improvements in **Recall, ICL** on the HELMET benchmark, as well as in **QA, Code, and Long-Dialogue History Understanding tasks** on LongBench V2.
>
>
> **Q3: "...Is it possible to do further scaling to observe more significant gains?"**
>
> **A3:** Due to our limited computational resources and time, we are unable to scale to larger data volumes within the current rebuttal period. However, as we discussed in **A5**, we compared our method with LongAlign and demonstrated that LongMagpie exhibits stronger scalability and more consistent performance improvements as the data scale increases.
>
>
>
> **Q4: "...This would clarify effectiveness at truly extreme versus moderately long contexts, and the effectivenes across tasks?"**
>
> **A4:**  As outlined in **A2**, we provide a task-based categorization, and the reported metrics there are averaged over context lengths of 8k, 16k, 32k, 64k, and 128k. To further clarify effectiveness under truly extreme context lengths, we present the detailed results for the 128k context length:
>
> | Method      | Recall       | RAG          | ICL          | Re-rank      | LongQA       | RULER        |
> |:-----------:|:------------:|:------------:|:------------:|:------------:|:------------:|:------------:|
> | ChatQA      | 82.69        | **61.88**      | 87           | 6.05         | **41.75**    | 78.64      |
> | LongAlign   | _87.75_      | 50.29        | 87.4       | 8.64       | 31.14        | 78.06        |
> | LongAlign + Ultrachat   |87.56 | _58.88_ | _89.60_ | _9.87_ | 33.60 | _82.05_ |
> | LongMagpie  | **95.44**    | 58.29    | **90**       | **14.55**    | _38.14_      | **84.11**    |
>
>
> These results show that LongMagpie continues to deliver leading or second-best performance even at extreme context lengths. We will provide a breakdown of results across different context lengths from 8k to 128k in the revised version.
>
>
> **Q5: "...could authors clarify this comparison or discuss if a sample-count-controlled comparison was considered?"**
>
> **A5:**  To ensure a sample-count-controlled comparison, we train on 190k samples across different methods. For ChatQA, we use its original 190k dataset;  For LongAlign, we follow its original construction strategy to generate a 190k version. Results are shown below:
>
> | Method      | Data Size | HELMET | RUELR | LongbenchV2 |
> |:-----------:|:------------:|:------------:|:------------:|:------------:|
> | ChatQA   | 190k      | 60.23  | _89.82_ | 30.8        |
> | LongAlign   | 190k      | _60.63_  | 87.36  | **33**        |
> | Longmagpie  | 190k      | **61.29**   | **90.65** | _32.6_        |
>
> We further scale up the LongAlign dataset to 450k samples to compare its scalability. Results are as follows:
>
> | Method      | Data Size | HELMET | RUELR | LongbenchV2 |
> |:-----------:|:------------:|:------------:|:------------:|:------------:|
> | LongAlign   | 450k      | 60.62   | 88.77 | 33.2        |
> | Longmagpie  | 450k      | **62.1**   | **91.17** | **34.4**        |
>
> As shown, LongMagpie consistently outperforms LongAlign on average, especially as the data scale increases. We attribute this to its ability to generate more diverse and higher-quality questions (as shown in Figure 2) through adaptive query generation, rather than relying on fixed prompt templates or seed questions.
>
> In addition, prior methods often depend on domain-specific long-context data or long-context-capable LLMs, which hinders their scalability. For example, ChatQA synthesizes data using NarrativeQA and needs to be combined with LongAlpaca12k and OpenOrca to reach 190k samples. LongAlign requires long documents and long-context models for data synthesis, and also needs to be mixed with short-text instruction data. In contrast, LongMagpie uses only general short-document datasets (~1.6k tokens average, as shown in Figure 3) and a simple, scalable method, enabling efficient synthesis at scale without external instruction data.
>
>
>
>
> **Q6: "How do authors plan to detect and mitigate biases inherited from the source LLMs in the generated data? "**
>
> **A6:** We plan to adopt the following strategies to detect and mitigate biases inherited from the source LLMs:
>
> * **Cross-model verification**: Inspired by the multi-agent debate framework proposed by Google [2], we will generate QA pairs for each document using multiple LLMs, then conduct a debate among them. A separate judge model is used to evaluate the debated outputs and select the most appropriate responses. This process helps reduce the likelihood of propagating biases or errors from any single model.
>
> * **Quality filtering**: We employ FsfairX-Llama3-RM-v0.1 to score and filter out low-quality generated data (As shown in Figure 2).
>
> * **Safety filtering**: In addition, we perform safety checks using Llama-Guard-3-8B. Our analysis shows that **99.86%** of the data passes the safety criteria. Detailed safety statistics are provided in our response to Reviewer KGZd (as we discussed in **A7**).
>
> **Q7: "...Additionally, can the framework be extended beyond document-query relationships to cover other critical long-context tasks (e.g., summarization, complex structured extraction, multi-turn dialogue)?"**
>
> **A7:** We categorize the generated QA pairs into various task types, as shown below:
>
> | Category                                | Count   |
> |-----------------------------------------|---------|
> | Precise Retrieval                       | 201,306 |
> | Summarization                           | 91,118  |
> | Advice Seeking                          | 51,609  |
> | Planning or Reasoning (Multi-step Analysis) | 38,526  |
> | Comparative or Choice-Based Task        | 25,342  |
> | Math or Data Analysis                   | 8,679   |
> | Complex Structured Extraction           | 5,725   |
> | Creative Task                           | 3,339   |
> | Coding & Debugging                      | 2,999   |
>
>
> As shown above, our framework already generates a substantial number of instances beyond traditional document-query pairs, including tasks related to summarization and complex structured extraction.
>
> In addition, the generated task types in LongMagpie often align closely with document content—e.g., code documents yield code-related queries, and structured texts lead to extraction tasks. We will expand to more diverse domains and formats to broaden task coverage, with concrete examples to be included in the revised version.
>
> While we do not explicitly construct multi-turn datasets, our method improves performance on LongBench V2's Long-dialogue History Understanding task (see **A2**), indicating implicit benefits for multi-turn dialogue.
>
> **Q8: "Any safely analysis of the generated dataset?"**
>
> **A8:**  Our analysis shows that **99.86%** of the data passes the safety criteria. Detailed safety statistics are provided in our response to Reviewer KGZd (as we discussed in **A7**).
>
>
> ---
>
> [1] Gao C, Wu X, Lin Z, et al. Nextlong: Toward effective long-context training without long documents. ICML 2025.
>
> [2] Du, Yilun, et al. Improving factuality and reasoning in language models through multiagent debate. ICML 2023.

---

> > ### Comment · Reviewer_iG7h · 2025-08-04
> >
> > I sincerely thank the authors for the detailed response. However, based on the new information provided, I suspect that it reinforces some of my original points about potential biases and task coverage.
> >
> > First, I checked the HELMET benchmark and realized that they contain 7 tasks, while the author only evaluated on 5 tasks. I found that there's no explanation in the paper nor in the response. Could you please clarify why the other two tasks, namely "Generation with citations" and "Summarization," were excluded from both the paper and this response? Can the authors provide scores for these two tasks?
> >
> > Second, in A7, it shows that "Precise Retrieval" tasks account for around half of the entire dataset. This significant skew towards retrieval-based questions seems to be a key factor driving the high performance on the Recall sub-task in HELMET and the overall RULER benchmark, which is also heavily focused on recall-style evaluation. This reinforces my original concern: the model excels at retrieval because the training data is overwhelmingly retrieval-focused, which may limit its generalization to other long-context skills. The less dominant gains on other tasks seem to support this. Given this significant data imbalance, how should we interpret the strong recall performance, and could you comment on this potential limitation?

---

> > > ### Author Response · Authors · 2025-08-06
> > > **Response to Reviewer iG7h**
> > >
> > > **Q9: "Could you please clarify why the other two tasks, namely "Generation with citations" and "Summarization," were excluded from both the paper and this response? Can the authors provide scores for these two tasks?"**
> > >
> > > **A9:** Our evaluation protocol follows the evaluation tasks used in [1]. We are willing to provide the results for the two tasks, and the full set of metrics is shown below:
> > >
> > >
> > > | Method      | Recall     | RAG       | ICL       | Re-rank   | LongQA    | Cite      | Summ.      | HELMET(Avg.) | HELMET(w/o Recall) | RULER     |
> > > |-------------|:------------:|:-----------:|:-----------:|:-----------:|:-----------:|:-----------:|:-----------:|:---------------:|:---------------------:|:-----------:|
> > > | ChatQA      | *93.34*    | **66.47** | 80.36   | 23.74     | **37.25** | 15.18     | 20.61     | *48.14*         | *40.60*             | *89.82*   |
> > > | LongAlign   | 92.43      | 59.05     | *81.20* | *27.12*   | 29.14     | *17.93*   | *24.32*   | 47.31       | 39.79               | 86.08     |
> > > | LongMagpie  | **97.53**  | *63.37*   | **85.84** | **28.60** | *35.16*   | **19.99** | **26.36** | **50.98**     | **43.22**           | **91.17** |
> > >
> > >
> > > As shown, our method achieves the best performance on both the Citations and Summarization tasks, further strengthening the overall advantage of our approach.
> > >
> > >
> > > **Q10: "the model excels at retrieval because the training data is overwhelmingly retrieval-focused, which may limit its generalization to other long-context skills."**
> > >
> > >
> > > **A10:** Based on our experimental results, we find that **retrieval-focused training data do NOT limit the model's generalization to other long-context skills. On the contrary, the improved retrieval capability facilitates performance across various tasks**. Intuitively, effective retrieval is a foundational skill for handling long-context inputs, as models must first identify relevant information before generating accurate responses. The importance of retrieval in long-context models has also been widely recognized in prior work [2][3].
> > >
> > > To investigate this more directly, we conduct additional experiments on the 190k-sample dataset. Specifically, we vary the proportion of Precise Retrieval data while adjusting the other data distributions accordingly. One setting reduces the Precise Retrieval portion from 50% to 30%, and the other increases it to 70%. The results are shown below:
> > >
> > > | Precise Retrieval (%) | Recall     | RAG       | ICL       | Re-rank   | LongQA    | Cite      | Summ.      | HELMET(Avg.) | HELMET(w/o Recall) | RULER     |
> > > |-------------|:------------:|:-----------:|:-----------:|:-----------:|:-----------:|:-----------:|:-----------:|:---------------:|:---------------------:|:-----------:|
> > > | 30%                    | *97.63*    | *62.99*     | 80.92     | 25.44     | 34.72     | 19.30     | **26.12** | 49.59         | 41.58               | *90.71*   |
> > > | 50%  | 97.29      | 62.72     | **85.12** | *26.26*   | *35.05*   | *20.39*   | 24.32     | *50.16*       | *42.31*             | 90.65     |
> > > | 70%                    | **98.85**  | **63.38** | *84.16*   | **26.85** | **36.26** | **22.02** | *24.86*   | **50.91**     | **42.92**           | **90.93** |
> > >
> > > As seen, **increasing the proportion of Precise Retrieval data improves the model's Recall performance, which also leads to consistent gains in downstream tasks such as RAG, Re-rank, LongQA, and Cite.** Overall, the average HELMET score improved by +0.75, and by +0.61 when excluding Recall, confirming that retrieval-centric training benefits general long-context capabilities. In addition, as shown in **A9**, LongMagpie also achieves better performance in the HELMET(w/o Recall) metric.
> > >
> > > More importantly, **we would like to emphasize the consistent performance improvements on non-retrieval tasks in LongBench V2**, as discussed in **A2**. LongBench V2 spans a wide range of realistic long-context scenarios and has been widely adopted in recent works such as Kimi K2 [4] and MiniMax-01 [5]. **Our method delivers substantial gains on LongBench V2 (+3.0, see Table 1), further supporting our view that retrieval-centered training enhances general long-context capabilities across diverse datasets**.
> > >
> > >
> > > We sincerely appreciate your thoughtful feedback. Please feel free to share any further questions or suggestions you might have.
> > >
> > > ---
> > >
> > > [1] Gao C, Wu X, Lin Z, et al. Nextlong: Toward effective long-context training without long documents. ICML 2025.
> > >
> > > [2] Fu, Yao, et al. Data engineering for scaling language models to 128k context. ICML 2024.
> > >
> > > [3] Wu, Wenhao, et al. Retrieval head mechanistically explains long-context factuality. ICLR 2025.
> > >
> > > [4] Team, Kimi, et al. Kimi K2: Open Agentic Intelligence. arXiv preprint arXiv:2507.20534 (2025).
> > >
> > > [5] Li, Aonian, et al. Minimax-01: Scaling foundation models with lightning attention. arXiv preprint arXiv:2501.08313 (2025).

---

> ### Comment · Reviewer_iG7h · 2025-08-06
>
> Thanks for the further experiments and analysis. I believe the authors have addressed most of my concerns. I have decided to raise my rating and recommend that the authors include the new results and discussions in the final version of the paper.

---

> ### Author Response · Authors · 2025-08-07
> **Response to Reviewer iG7h**
>
> We sincerely thank you once again for your thoughtful and constructive feedback. We are glad to hear that your concerns have been addressed. **All results and discussions from the rebuttal period will be incorporated into the final version of the paper.**

---

### Comment · Area_Chair_357h · 2025-08-09
**Reviewers' Mandatory Acknowledgement**

Hi all, thanks for the active discussion and engagement! All reviewers agree the work is in good shape, and the authors have addressed most of the concerns from the initial reviews.

For reviewers who have not yet added the Mandatory Acknowledgement, please do so as soon as possible — less than 24 hours remain in the discussion period. Thank you!

---

### Decision · Program_Chairs · 2025-09-17

**Decision:**

Accept (poster)

**Comment:**

This submission proposes a self-synthesis framework LongMagpie for generating large-scale, high-quality long-context instruction data without human annotation or predefined templates. The approach leverages the auto-regressive capabilities of aligned LLMs: when presented with a document and special tokens indicating a user query, the LLM generates contextually relevant queries and responses, producing document-query-response triplets at scale. LongMagpie also introduces a probabilistic mixing strategy p-Mix combining long- and short-context instructions to balance performance across tasks. Experiments on HELMET, RULER, and LongBench-v2 demonstrate that the proposed method achieves sota results on long-context tasks while maintaining competitive short-context performance.

Strengths of the submission:

1. self-synthesis attempt (iG7h, KGZd): the authors introduces a simple and effective approach that leverages aligned LLMs to generate long-context instructions automatically.
2. simplicity and clarity of approach (vum6): the proposed framework is conceptually straightforward and well-presented, making it easy to adopt.

Weaknesses pointed out by reviewers:

1. bias inheritance and limited task coverage (iG7h, KGZd): since the proposed method relies on aligned LLMs for data generation, any existing biases or limitations in the base models are likely propagated. in addition, the current focus on document-query-response triplets restricts its applicability to broader long-context tasks.
2. modest improvements and scalability concerns (iG7h, vum6): on some benchmarks, performance gains over strong baselines are marginal, and ablation studies show diminishing returns when scaling up the dataset size. The impact of the p-Mix strategy on balancing long- and short-context performance is also limited. Reviewers also noted that comparisons across datasets and benchmarks could be more transparent, including detailed breakdowns by task type and context length, as well as clarifications around dataset sizes and training configurations.

Overall, the submission presents a technically sound and practical approach to an important problem. While there are valid concerns regarding scalability, bias inheritance, and evaluation clarity, the method’s simplicity, and empirical improvements make it a valuable contribution to the field. During the rebuttal, the authors managed to address most issues, making it with all positive ratings. All factors considered, this submission at its current format is given a positive recommendation.